# Malnutrition and Allergies: Tipping the Immune Balance towards Health

**DOI:** 10.3390/jcm13164713

**Published:** 2024-08-11

**Authors:** Emilia Vassilopoulou, Carina Venter, Franziska Roth-Walter

**Affiliations:** 1Department of Nutritional Sciences and Dietetics, School of Health Sciences, International Hellenic University, 57400 Thessaloniki, Greece; 2Department of Clinical Sciences and Community Health, Univertià degli Studi die Milano, 20122 Milan, Italy; 3Pediatrics, Section of Allergy & Immunology, University of Colorado Denver School of Medicine, Children’s Hospital Colorado, Box B518, Anschutz Medical Campus, Aurora, CO 80045, USA; 4Messerli Research Institute, Department of Interdisciplinary Life Sciences, University of Veterinary Medicine Vienna, Medical University of Vienna and University of Vienna, 1210 Vienna, Austria; 5Institute of Pathophysiology and Allergy Research, Center of Pathophysiology, Infectiology and Immunology, Medical University of Vienna, 1090 Vienna, Austria

**Keywords:** allergens, iron, vitamin A, nutritional immunity, malnutrition, malabsorption, Th2, micronutrient deficiencies, protein deficiencies, type 2 inflammation

## Abstract

Malnutrition, which includes macro- and micronutrient deficiencies, is common in individuals with allergic dermatitis, food allergies, rhinitis, and asthma. Prolonged deficiencies of proteins, minerals, and vitamins promote Th2 inflammation, setting the stage for allergic sensitization. Consequently, malnutrition, which includes micronutrient deficiencies, fosters the development of allergies, while an adequate supply of micronutrients promotes immune cells with regulatory and tolerogenic phenotypes. As protein and micronutrient deficiencies mimic an infection, the body’s innate response limits access to these nutrients by reducing their dietary absorption. This review highlights our current understanding of the physiological functions of allergenic proteins, iron, and vitamin A, particularly regarding their reduced bioavailability under inflamed conditions, necessitating different dietary approaches to improve their absorption. Additionally, the role of most allergens as nutrient binders and their involvement in nutritional immunity will be briefly summarized. Their ability to bind nutrients and their close association with immune cells can trigger exaggerated immune responses and allergies in individuals with deficiencies. However, in nutrient-rich conditions, these allergens can also provide nutrients to immune cells and promote health.

An often overlooked aspect in the etiology of allergies is the fact that not everyone becomes allergic despite being exposed to the same allergenic and environmental threats. This suggests that certain intrinsic conditions promote allergies, making an individual particularly sensitive to allergenic substances. The term “atopy” reflects the hypersensitive predisposition of a person prone to allergies compared to a person with no such risk.

Here, we review the diet as a major contributor to “atopy”, highlighting how deficiencies in specific proteins, minerals, and vitamins can shift the immune system toward type 2 inflammation. We focus on three common deficiencies worldwide, protein, iron, and vitamin A, and explore their impact on ameliorating the disease course.

## 1. Definition of Malnutrition

Though malnutrition often refers to undernutrition, encompassing protein–energy malnutrition and micronutrient deficiencies, it also includes overnutrition, referring to both deficiencies and excesses in nutrient intake. Obese individuals are often deficient in micronutrients despite calorie excess [1]. Moreover, the World Health Organization (WHO) also includes impaired nutrient utilization as one form of malnutrition.

Malnutrition is not always the result of an inadequate intake of micro- and macronutrients but may also arise from impaired nutrient utilization. While micronutrient deficiencies are more frequent and severe among disadvantaged populations, they also represent a public health problem in industrialized countries [2]. The NHANES study revealed inadequate intake of vitamins and minerals in about 40% of Americans [3,4,5], highlighting that malnutrition is a problem in industrialized, developed countries. About 40% [6] of all people with chronic illnesses and up to 90% of hospitalized people suffer from malnutrition [7], which is an independent risk factor for increased morbidity and mortality. Reasons for hospital-acquired malnutritionis beside poor appetite hospital meal refusal, operation-related fasting, polypharmacy, and comorbidities [7], with malnutrition being a predictor of postoperative morbidity and mortality [8]. The increased consumption of highly processed, energy-dense but micronutrient-poor foods in industrialized countries, and increasingly in those undergoing social and economic transition, adversely affects micronutrient intake and status [9]. Certain lifestyle factors also increase the risk of malnutrition, such as frequent blood donations, smoking, vegan/vegetarian diets, excessive use of some medication (e.g., antacids hindering dietary iron uptake), and greater nutritional demands during specific life stages (growth, pregnancy, or endurance sports) [10].

### 1.1. Malabsorption Due to Inflammation and Nutritional Immunity

Both inflammation and malnutrition can evoke “nutritional immunity” [11], one of the most evolutionary conserved innate mechanisms present in all living organisms. In humans, nutritional immunity involves the redistribution of vitamins and minerals away from circulation [11,12] to safer havens, such as the macrophages and liver cells, while it also hinders the dietary uptake of micronutrients. Consequently, “nutritional immunity” impairs nutrient utilization within the body and results in malabsorption, with “impaired nutrient utilization” listed by the WHO as one form of malnutrition.

People at risk of nutritional deficiencies include obese people, due to low-grade inflammation, individuals with inadequate nutrient intake, and those suffering from chronic diseases including congestive heart failure [13,14,15], chronic kidney diseases [16,17,18] autoimmune diseases [19,20,21,22,23], inflammatory bowel disease [24,25], cancer [26,27,28], and allergic diseases [29,30,31,32,33] (Figure 1).

### 1.2. Physiological Pathways of Nutrient Absorption

Nutrient absorption is a complex process that depends on many variables, including the interinfluence among them and one’s overall nutritional status [34]. Most nutrients are absorbed in the jejunum, while B12 and bile salts are absorbed in the terminal ileum. Iron and most minerals [35] are absorbed in the duodenum, while magnesium’s predominant absorption site is the terminal ileum and proximal colon [36]. Once nutrients pass the gut lining, they enter either directly into the bloodstream or via the lymphatic system, before being released into circulation (Figure 2).

Micronutrients are assimilated via three major mechanisms:(1)Direct bloodstream absorption: Simple sugars [37] (glucose, fructose), amino acids (from digested proteins), short-chain fatty acids SCFA and medium-chain fatty acids MCFA, water-soluble vitamin C, vitamin Bs (except B12) and folate [38], and most minerals (iron, calcium, potassium) are readily absorbed into the bloodstream and enter via the portal vein the liver.(2)Lymphatic absorption: Fat and fat-soluble compounds (including fat-soluble vitamin A, D, E, and K) are taken up via the lymphatic vessels [39]. These nutrients are emulsified in micelles composed of bile salts absorbed by the enterocytes and packed into chylomicrons [40,41]. Chylomicrons are composed primarily of triglycerides, phospholipids, cholesterol, and lipoproteins and enter the bloodstream near the collarbone.(3)Receptor-specific uptake: Digestion-resistant proteins serve as carriers for nutrients (iron [2,42,43,44,45,46,47,48,49,50,51], calcium/magnesium [52,53], vitamins [43,44,47], carbohydrates [54,55], phenolics [48,49,50,56], and lipids [57,58,59]). This receptor-specific uptake occurs often, typically exploiting the lacteals—the jejunal lymphatic vessels [60]—as shown with the absorption of milk proteins (whey) [61] and egg proteins [62,63] but also for plant-derived food, such as soy [64,65] and nut proteins [64,65].

Notably, these nutrient-binding proteins are sensitive to heat and food processing, which can impair their nutrient-carrying abilities. Minerals such as iron, zinc, selenium, and calcium have reduced bioavailability in inflamed settings via a range of mechanisms, including the presence of a mucosal block for iron [66,67] and zinc [68,69], a compromised epithelial barrier function affecting e.g., folate [70], as well as vitamin B12 [71,72,73] absorption. Iron and vitamin D deficiencies can affect calcium and magnesium [74] absorption, while vitamin B6 deficiency increases zinc and lowers copper but not iron absorption [75]. Also, uptake of fat-solubilized vitamin A [76] can be compromised in an inflamed setting, likely due to an altered bile secretion profile [77], and is exacerbated by zinc and iron deficiency, with the bioavailability of vitamin D also being reduced in inflamed settings. A lack of iron [78,79,80] and vitamin A [81,82,83] has detrimental health consequences and is associated with increased morbidity and mortality [84,85].

## 2. General Immunological Implications of Malnutrition (Protein and Micronutrients): A Shift towards Th2

### 2.1. Malnutrition and the Thymus

Particularly, protein and micronutrient deficiencies in minerals (iron, zinc, magnesium) or vitamins (A, Bs, C, D) have a direct impact on our immune system and drive inflammation [11] via their effect on the thymus and other lymphoid organs. A lack of these proteins, vitamins, and minerals causes atrophy of lymphoid tissues, and here, the thymus and the lymph organs are the major organs particularly affected. The thymus was described as a very sensitive barometer of malnutrition by Menkel 200 years ago [86,87], when its function was not clear. The thymus is, beside the bone marrow, a primary lymphoid organ, which is located in the upper chest area and is essential for lymphocyte production (though it also contains macrophages, dendritic cells, and a small number of B cells, neutrophils, and eosinophils). It is very sensitive to nutritional deprivation resulting in the depletion of immature CD4+CD8+ cells [88,89] and is associated with a Th2 skewing [90].

Nutrient deficiencies induce similar cellular changes in the thymus, such as acute infections. Already, mild malnutrition [88,91] induces the following changes in the thymus: An increase in the extracellular matrix, which contains fibronectin, laminin, and type IV collagen (preceding thymocyte depletion), followed by thymocytic depletion. This goes along with a relative rise in macrophages (phagocyting dead thymocytes) and less production of thymulin-a zinc-containing thymic peptide hormones capable of downregulating inflammatory mediators [89,92,93].

The depletion of thymocytes [89] is a consistent finding in individuals consuming a diet deficient in proteins for a prolonged period (in *in vivo* models, deficits are seen from two weeks on, despite calorie sufficiency) [94,95,96,97] but also when single nutrients, such as iron [98,99], zinc [100], magnesium [101,102,103], or vitamin (B1/B2/B5/B6/C) [104,105], are missing. Regarding the type of immune response generated during malnutrition, the skewing of the immune system towards Th2 is well established [106] and already seen upon “moderate malnutrition”. Moderate malnutrition is defined after the Global Leadership Initiative on Malnutrition GLIM’s definition, with a weight loss of 5–10% within the past 6 months, BMI 18.5–20 m/kg^2^ (age < 70 years), and/or BMI < 22 m/kg^2^ (≥70 years). Severe malnutrition includes weight loss >10% within the past 6 months, BMI <18.5 m/kg^2^ (<70 years), and/or BMI < 20 m/kg^2^ (≥70 years) [107]. Importantly, thymic atrophy is reversible upon providing the missing nutrients.

One cannot overemphasize that nutritional deficits not only can elicit inflammation [108] but also lay the groundwork for a Th2 environment (typically characterized by the presence of IL4, IL13, and/or IL5), which is the prerequisite for allergic sensitization. In the acute phase, nutrient deficiencies usually manifest as a Th1/Th17 response, before this changes to a Th2 response in the chronic phase [11,109]. Immunological changes are reported in apparently healthy people with a low BMI (<18.5) showing subclinical inflammation with elevated Th1 (IL2, IL12, TNFα,) and Th2 cytokines (IL4, IL5, IL13, IL10) [110]. Malnourished children also show greater IL4 and IL10 levels [111], while asthma risk seems to increase [112]. IL4 concentrations are significantly higher in school children from Tanzania affected with stunting [113]. Thus, persistent malnutrition *per se* results in the establishment of a Th2 milieu [90,112,114,115,116,117,118]. Importantly, these findings are consistently reproduced in clinical settings, with inferior intake of several minerals and vitamins correlating with higher IL4 and IL10 levels.

### 2.2. The Importance of Allergenic Proteins: Saviors or Dangers

Proteins are essential for human health, as inadequate protein consumption leads to type 2 inflammation. Animal studies consistently demonstrate that a protein-poor diet results in low-grade inflammation (IL4/IFNγ) [119], while increased dietary proteins strongly correlate with the immune system’s abilities to combat parasites [120]. Mice fed amino acids instead of proteins had poorly developed gut-associated lymphoid tissue [121] and normal IgM but reduced circulating IgG and IgA, and they showed a predominant Th2 profile [122]. In pigs, a high protein diet results in an improved barrier in the ileum and decreased glucose and glutamine transport [123]. Protein–energy malnutrition is an important cause of secondary immune deficiency and is associated with high TNFα, IFNγ, and IL4 levels in moderate/severely malnourished children [124] and occurs in the [125] Western world usually in the context of chronic diseases [126].

While most proteins are harmless, a few protein families tend to trigger allergic reactions. Indeed, these allergens are usually clustered in specific protein families (Table 1). In mammals, the most important major allergens often belong to the “lipocalin” superfamily [66]. Allergens from plants originate from protein families that are often seed proteins, including the pathogenesis-related proteins PR10 family; the prolamin superfamily (including seed storage protein 2S albumins and the nonspecific lipid transfer proteins nsLTPs) [127,128,129]; the cupin protein superfamily (including the legumins and vicilins); and Ole e 1 families [130].

A key feature of most major allergens is their ability to bind to, transport, and enhance the absorption of essential nutrients. This ability has been observed for iron (in lipocalins, seed storage proteins 7S, 11S, and PR10 proteins, Alt a 1, 2S albumin) [2,42,43,44,45,46,47,48,49,50,51,56,131], zinc [132,133,134,135,136] (lipocalin, Ole e 1, nsLTPs), lipids (in lipocalins, PR10 and LTPs) [132,137,138], and vitamin A (lipocalins, PR10, Alt a 1) [43,44,45,46,47,121]. Many plant proteins can bind iron via antioxidative flavonoids, with the ligands for PR10, 2S protein, 7S proteins identified in birch and hazelnut having quercetin as a common core structure [139,140], strawberry proteins binding to catechins [141], peanut proteins Ara h 2 and Ara h 6 capable of binding to the flavonoid epigallocatechin-3-gallate [142], quercetin, [143] epicatechin [144], or proanthocyanidins [145], with binding decreasing their allergenicity [146].

Interestingly, studies suggest that iron binding may play a role in reducing the immunogenicity of these proteins, hence their ability to trigger an immune response. This has been demonstrated with peanut allergens (Ara h 1–7S, Ara h 3 -11S protein) [51], egg proteins (Gal d 3 [147]), lipocalin beta-lactoglobulin (Bos d 5) [48,50], and the birch allergen Bet v 1 [49,56].

Food processing often changes mineral and vitamin content and alters the structures of these proteins. For example, the pasteurization of milk promotes the aggregation of whey proteins [61] and impairs the ligand-binding capacity of the whey protein beta-lactoglobulin, shown with ligands such as retinol and palmitic acid [148], while simultaneously increasing its antigenicity [61,148]. Similarly, pasteurization decreases copper and iron content [149] in milk.

Studies by us and others suggest that their nutrient-binding features switch these proteins from tolerogenic (with nutrients) to allergenic (without nutrients). Nutrient-poor conditions turn birch, peanut, egg proteins, and milk proteins into potent allergens, while micronutrient-adequate conditions appear to promote immune resilience [30,42,43,44,48,49,50,56,131].

Additionally, the function of these proteins should be considered, as the majority of these protein families are known to act during the stress response and nutritional immunity in their respective plants [150,151] and organisms [2,42,48,66,131,150,152,153,154,155,156,157,158,159,160,161,162,163,164].

These findings support an emerging principle that these proteins can deprive their local environment of important nutrients such as iron, lipids, or vitamins and thereby have a profound metabolic impact on our immune cells. This depletion triggers a danger signal to the immune system, especially in atopic individuals. If, on the other hand, nutrients are available in abundance, these same proteins act as carriers for micronutrients. They bind and deliver essential nutrients (become “holo-proteins”) and contribute to immune cell health by promoting tolerance.

Essentially, the allergenicity of these proteins appears to be a context-dependent phenomenon. Their function can change depending on nutrient availability, from useful nutrient carriers in a nutrient-rich environment to potential allergens under nutrient-poor conditions.

**Table 1 jcm-13-04713-t001:** Summary of allergenic protein families on known ligands and biological function.

Protein Families	Structure	Examples	Known Ligands	Origin	Function	Ref
Pathogenesis-related class seed storage protein	small protein with antiparallel beta-strands and alpha-helices	Bet v 1, Pru a 1, Mal d 1, Fra a 1	phytohormones, siderophores, flavonoids, alkaloids	plants	Pathogenesis-related proteins (PRPs): signature genes for systemic acquired resistance in plants Microbicidic, Kunitz type of protease inhibitor	[154,155,156,157,158,159,160]
Ole e 1 Family	β-barrel fold, stabilized by 3 disulfide bond, heat stable	Ole e 1, Pla l 1, Che a 1	2+ metals	plants	Pollen tube development, leave senescence Activated under ROS induction, contribute to antioxidant production, plant defense responses.	[135,161,162,163,164]
nsLTPs seed storage proteins Prolamin superfamily	cysteine-rich alpha-helical; rich in proline and glutamine	Pru p 3, Ara h 9, Fra a 3	Fatty acids, phospholipids	plants	Antimicrobial, lipid utilization, plant stress Regulate FAO, binds to calmodulin (central hub in calcium-dependent cellular regulation)	[154,165]
2S albumin (conglutin) seed storage proteins prolamin superfamily	small cysteine-rich, alpha-helical protein	Ara h 2, Ber e 1, Ses I 1, Gly m 8	phenolics	plants	Nutrient reservoir, regulate germination Antimicrobial, stress response [166]	[154,165]
Albumin 2S seed storage	hemopexin-like fold no disulphide bonds thermostabile, ß-propeller		heme, spermidine thiamine	plant	Stress response, antioxidative, agglutinate erythrocytes; peroxidase activity or heme binding Seed germination	[167,168,169]
cereal prolamins prolamin superfamily	alpha-helical, conserved cystein-skeleton; rich in proline and glutamine	Tri a 19 (wheat), Sec c 20 Hor v 21	copper, sugars, fats, phenolics	plants	Nutrient reservoir, regulate germination Antimicrobial, stress response	[166,170]
prolamin alpha-amylase inhibitors [171]	alpha-helical, cystein-rich	Tri a 28 Hor v 15	calcium	plants	Antimicrobial, stress response Inhibit exogenous insect amylases	[172]
7S/vicilin CUPIN	beta-barrel core [165]	Ara h 1, Jug r 2, Ses I 3	copper, sugars, fats, phenolics	plants	Nutrient reservoir, regulate germination, Antimicrobial, stress response	[154,165,166]
11S/legumin-like CUPIN	beta-barrel core	Ara h 3, Ber e 2, Ses i 6	copper, sugars, fats, phenolics	plants	Nutrient reservoir, regulate germination Antimicrobial, stress response [166]	[154,165]
Lipocalins	symmetrical β-barrel fold,	Can f 1, Fel d 4, Bos d 5	siderophores, phenolics, vitamin, heme products	animal	Stress response, microbicidic, nutritional immunity	[66,173,174]
Serum Albumin	globular, several long α helices	Fel d 2, Gal d 5, Can f 3, Equ c 3	Cu^2+^, Zn^2+^, hormones, vitamins, minerals, drugs, hemin	animal	Carrier protein, nutritional immunity Negative acute phase protein Anti-inflammatory	[175,176,177]
Parvalbumin	calcium-binding, long α helices, EF-hand superfamily	Cyp c 1, Gad c 1	Ca^2+^, phosphatidylcholine, phospatidylethanolamine	animal	Calcium buffer, immunomodulatory Protective against reactive oxygen species, antibacterial	[178,179]
Tropomyosin	two-chained, α-helical coiled coil protein	Bla g 7, Lep s 1, Der f 10	actin	animal	Regulates stress fiber assembly Regulatea calcium-dependent interaction of actin/myosin during muscle contraction Host defense, immunomodulatory	[180,181,182]
Uteroglobin	homodimeric, alpha helical strucure linked by disulfide bonde	Fel d 1, Ory c 3	phosphatidylcholine, phosphatidylinositol, polychlorinated, steroids, environmental toxins progesterone	animal	Anti-inflammatory, antioxidant Inhibitor of phospholipase A2 Increased vulnerability to oxygen toxicity in uteroglobin-knock-out mice, defects in uteroglobin are associated with a susceptibility to asthma; protects epithelial linings	[183,184]
NPC2 proteins MD-2-related lipid recognition family	immunoglobulin-like β-sandwich fold	Der p 2, Gly d 2, Tyr p 2	lipids cholesterol other sterols, LPS	animal	Crucial for cholesterol transport and utilization	
Arginin Kinase	mainly α-helical	Bla g 9, Pen m 2, Der p 20	ATP and L-arginine phosphoarginine	animal	Immunomodulatory, stress response Storage of phosphoarginine Cell signaling, apoptosis	[185,186,187]

### 2.3. Iron Deficiency

Iron deficiency is the most common deficiency worldwide and can lead to mucosal inflammation, a weak immune system, anemia, cognitive deficits in children [188], preterm birth, low birth weight [189], and increased mortality [78,79,80,190]. Even in healthy adults, iron deficiency is a driver of low-grade chronic inflammation [191]. Notably, functional iron deficiency has been linked to the highest risk for mortality in chronic kidney diseases [192].

There are different terms used to define a suboptimal iron status:Iron deficiency/functional iron deficiency

Due to a lack of international consensus, iron deficiency is often defined as (1) serum ferritin < 100 ng/mL, or 100–299 ng/mL with transferrin saturation <20%, which is the guideline definition for heart failure patients, (2) serum iron concentration ≤ 13 μmol/L, or (3) transferrin saturation < 20% [193]. Iron deficiency, the most common nutrient deficiency globally, is associated with increased mortality, even in seemingly healthy populations [194]. Children under five years old, adolescents, and women of childbearing age are particularly at risk.
Anemia/absolute iron deficiency

It is important to note that iron deficiency is not synonymous with anemia. Iron deficient anemia, also known as absolute iron deficiency, is an extreme form of iron deficiency characterized by a measurable lack of hemoglobin in red blood cells, which are the most abundant cells in the human body, making up over 80% of all body cells. Absolute iron deficiency is defined by severely reduced or absent iron stores, whereas functional iron deficiency involves adequate iron stores but insufficient iron availability for incorporation into erythroid precursors. This often results from immune activation and the retention of iron in macrophages, the central hub for iron distribution and recycling in the human body [195]. Markers such as ferritin, typically used to assess body iron, are elevated under inflammatory conditions due to their role in the acute phase response, masking the presence of iron deficiency. In 2022, the global pooled prevalence of iron deficiency anemia was 16%, with 18% experiencing iron deficiency without anemia [196]. 

#### 2.3.1. Iron Deficiency Shifts the System toward Th2

Iron deficiency per se is associated with low-grade inflammation [197,198,199,200] (measured by low serum iron levels, low transferrin saturation, and elevated high-sensitivity C-reactive protein, alpha1-acid glycoprotein CRP) [201], and more proinflammatory monocytes in children [202] and infants [203]. Initially, inflammatory mediators like IL6 and TNFα are induced, which then shifts toward elevated IL4 levels [198,199,200] in more severe iron-deficient cases in children. Th1 cells, compared with Th2 cells, are particularly sensitive to iron deprivation [204,205] (Figure 3).

#### 2.3.2. Cell-Specific Alterations under Iron Deficiency

##### Macrophages

In humans, macrophages are not only central in the defense against pathogens, clearance of senescent cells, and wound healing. They also represent the central hub for iron distribution [206]. About 20–30 mg of iron is recycled daily from senescent red blood cells by splenic macrophages, whereas only 1–2 mg of dietary iron is absorbed daily [174].

Iron is a key regulator for immune function. Primarily, regulatory M2 macrophages can take up, recycle, and distribute iron. These cells are characterized by a large cytosolic iron pool, also known as the labile iron pool, low ferritin levels, and high expression of iron export/import proteins. The typical M2 marker is CD163, the hemoglobin/haptoglobin receptor, which contributes to iron homeostasis [207]. In contrast, M1 macrophages have a low labile iron pool and high levels of ferritin, in which iron is hidden from potential pathogenic invaders. M1 macrophages do not distribute iron, and under chronic inflammatory conditions, iron is retained by splenic macrophages resulting in anemia of chronic disease [208,209,210].

A chronic lack of iron or immune activation will change the by-default regulatory phenotype of macrophages, as iron turnover and the labile iron pool decrease [66]. The mitochondrial metabolic function, which heavily relies on iron, is impaired (citrate cycle and oxidative phosphorylation), causing a metabolic switch towards anaerobic glycolysis and increasing glucose uptake [211]. As such, iron deficiency changes the phenotype of macrophages, causing them to acquire characteristics of inflammatory cells. Infants [203] and children [202] with iron deficiency have monocytes with a proinflammatory signature, while a large labile iron pool is associated with an immature, regulatory macrophage phenotype [48,56].

Macrophages/monocytes [212,213], neutrophils, and NK cells [214] need iron for microbial killing, where they act as a catalyst for the generation of reactive oxygen species (ROS) [212,213,214]. A lack of bioavailable iron thus also impedes pathogen elimination, as the local and precise generation of ROS is impaired [215]. This is despite the greater “inflammatory” but ineffective activity that may contribute to the characteristics of senescent cells [216].

##### T Cells

An important aspect of iron deficiency is that lower red blood cell values often go along with an expansion of white blood cells. The lymphocytic population is elevated [217]; despite that, CD4+ cells and the CD4/CD8 ratio under iron-deficient conditions are reduced [217,218]. Iron deficiency or iron chelation impairs T-cell proliferation and results in apoptosis of proliferating activated T lymphocytes but not of resting peripheral blood lymphocytes or granulocytes [219]. In contrast, when sufficient iron is present, Th1, Th2, and Th17 differentiation [220] is repressed. Interestingly, T lymphocytes also partake in iron homeostasis, as T cell deficiency results in iron accumulation in the liver and pancreas [221]. Particularly, Th1 cells are very sensitive to iron-deficient conditions as the IFN-gamma/STAT1 signaling pathway is regulated by iron [204,222]. In contrast, Th2 cells are more resistant under iron-poor conditions, resulting in a shift toward a Th2 response and an increase in the cytokine IL-4 in humans with iron deficiency [198,199,200].

##### IgE Antibodies

In B cells, iron deficiency activates activation-induced cytidine deaminase (AID), an enzyme responsible for class switching and the affinity maturation of antibodies [223]. The lack of iron hampers heme synthesis in the mitochondria and maintains Bach2 activation [224] in B cells. Iron fortification studies significantly improved hemoglobin and serum ferritin levels, but also resulted in decreasing total IgE levels [225] in children and women [226]. Increased IgE levels are also commonly observed, such as sickle cell anemia [227] and autoimmune hemolytic anemia (where antibodies attack red blood cells [228]), upon infections [229] such as plasmodium falciparum malaria digesting hemoglobin in red blood cells, leading to anemia [230].

##### Epithelial Cells and Hair

Iron deficiency can disrupt the tight junctions in the gut epithelium, leading to increased permeability [231]. It also contributes to hair loss in Tmprss6 mask mice (with a defect in iron sensing), with fur regrowing with a high-iron diet [232]. An iron-restricted diet results in hair loss in IL10-deficient mice [233]. In humans, iron deficiency has been suggested as a contributor to nonscarring alopecia [234], telogen effluvium [235], and androgenic alopecia [236].

##### Mast Cells and Eosinophils

Lastly, mast cells are primed under iron-deficient conditions. The intradermal application of the iron binder desferrioxamine can activate connective-tissue-type mast cells and has been suggested as a positive control in intradermal skin tests. This induces a local iron deficiency, a concentration-dependent histamine release [237], and wheal formation, both *in vitro* and *in vivo* [237]. Conversely, the activation of mast cells can be hampered by the addition of iron-containing proteins such as transferrin, lactoferrin, and the iron-loaded whey protein beta-lactoglobulin [238,239,240,241,242]. Iron deficiency has been reported to cause and increase the prevalence of chronic generalized pruritus [243] and contribute to uremic pruritis in patients with chronic kidney diseases. *Vice versa*, in a clinical study, oral iron supplementation for 2 months was able to ameliorate chronic idiopathic urticaria in all 81 patients with mild hyposideremia [244]. The rare but described symptoms of anemia rashes in people afflicted with iron-deficient or aplastic anemia may further hint toward the priming of mast cells under iron deficiency. Eosinophils also seem to be promoted under iron-deficient conditions and repressed under iron-sufficient conditions in a murine model of allergic asthma [245]. In asthmatics, serum iron is negatively correlated with eosinophil counts [246], and poor fetal iron is speculated to be a risk factor for infant eosinophilia [247].

Thus, the extent of iron repletion in mast cells determines their priming state to release mediators such as histamine that are responsible for allergy symptoms, and eosinophils seem to be promoted under iron deficiency as well (Figure 3).

### 2.4. Vitamin A Deficiency

Vitamin A deficiency is the second most common deficiency worldwide. The WHO estimates that about 250 million preschool-aged children globally have subclinical or clinically relevant low serum vitamin A levels [248,249]. Importantly, vitamin A supplementation has repeatedly been shown to reduce “all-cause mortality” [81,82,83] and is also linked with iron homeostasis. Vitamin A supplementation alone can improve hemoglobin levels [250]. Vitamin A deficiency results in the impairment of vision, epithelial integrity, and inflammation [45,251], manifesting in its extreme forms in xerophthalmia and night blindness. Even subclinical vitamin A deficiency is associated with inflammation, iron deficiency, and increased all-cause morbidity [252,253]. During infection or inflammation, serum retinol levels decline [254,255], complicating the accurate assessment of vitamin A status when based solely on serum retinol levels due to concurrent rises in C-reactive protein [256].

#### 2.4.1. Vitamin A Deficiency Results in Type 2 Inflammation

Vitamin A is essential for the mucosal immune system and the epithelial barrier, regulating the transcription of many genes. In apparently healthy people, low retinol levels are associated with elevated CRP levels [257].

#### 2.4.2. Cell-Specific Alterations under Vitamin A Deficiency

##### Macrophages

In macrophages, vitamin-A-rich conditions suppress macrophage activation, differentiation [258], and inflammation cascade [259,260,261], while promoting a regulatory phenotype via STAT6 [262]. Similarly, IL4 can induce retinol production and excretion in macrophages via STAT6 [263].

##### Lymphoid Cells

As with iron, acute vitamin A deficiency will initially result in a Th1 response and IFNγ production [264,265], with a shift toward Th2 in the chronic phase [266,267], leading to elevated IgE levels *in vivo* [266]. In contrast, conditions with sufficient retinoic acid promote innate lymphoid cell ILC3s and T-regulatory cells [268]. A lack of vitamin A also results in a dramatic expansion of IL13-producing innate lymphoid cells [267,269].

##### Epithelial Cells

Vitamin A is essential for the skin and gut barrier function. Topical retinol application improves epithelial cell integrity and filaggrin expression [270] in UV-damaged skin, while retinoic acid intake improves intestinal epithelial cell differentiation and barrier function [271]. Vitamin A deficiency promotes squamous epithelial cell differentiation [272,273] and hyperkeratosis [274], which can be corrected by vitamin A supplementation *in vivo* [275]. Similarly, lung epithelial cell proliferation is suppressed by vitamin A intake [276].

##### Mast Cells

Vitamin A deficiency potentiates mast cell activation [277], while retinol absorption improves atopic dermatitis symptoms [278]. A concentration-dependent stabilizing effect on mast cells and histamine release has been reported *in vitro* [279] and *in vivo* [45,280,281] (Figure 3).

## 3. Malnutrition in Allergic Diseases

Both a high as well as a low body mass index (BMI) are associated with allergic diseases. As such, a high BMI contributes to disability-adjusted life years and death in asthma [282,283], and a low BMI is associated with allergic sensitization [284], food allergies [285,286,287,288], and allergic rhinitis [32,289].

Several studies have reported reduced intake of macronutrients, particularly proteins [290], and micronutrient deficiencies [285], such as vitamin A [291] and iron [285,287], in populations with food allergies. Interestingly, cow milk allergy is associated with malnutrition [286,288], with a large retrospective study confirming that children with cow milk allergies are significantly shorter and weigh less than nonallergic children [292]. Food restriction in children with atopic dermatitis and/or food allergy was also linked with stunting, underweight, and increased disease severity [293,294].

### 3.1. Iron Deficiency/Anemia and Atopic Diseases

Large epidemiological studies in the US [295], Korea [296,297], and Japan [298] consistently reported that people [295,296,297,299] with atopic diseases [295,296] are much more likely to be anemic—and lack iron—compared with those without any allergy. Children with atopic dermatitis [300] are more likely to have iron and zinc deficiencies [299], and low serum iron is associated with lower lung function [301]. Among food allergies, cow milk allergies are particularly a risk factor for iron-deficient anemia [302,303,304,305]. Additionally, maternal iron status during pregnancy affects children’s health outcomes. A lower iron status during pregnancy is associated with childhood wheezing, decreased lung function, and allergic sensitization [306,307,308,309]. Low cord blood iron levels at birth are associated with atopic urticaria, infantile eosinophilia, and wheezing by age four [247,306]. Conversely, a good iron status during pregnancy lowers the risk of developing atopic dermatitis [247] and asthma [247,306,307,309,310,311] in children. Low serum iron levels are inversely related to blood eosinophil counts in asthmatic adults [246], with adequate iron stores decreasing the odds of lifetime asthma, current asthma, and asthma attacks/episodes [312].

Allergic diseases are also more common in patients with anemic diseases, in which elevated IgE is commonly observed and not related to parasitic infestations [225]. Patients with beta-thalassemia major (Cooley’s anemia), who develop chronic anemia due to impaired hemoglobin synthesis and possess often enlarged spleens, livers, and hearts, are more likely to have allergic diseases [313,314] and suffer from asthma [313,315,316,317]. Similarly, also atopic dermatitis subjects have a greater risk of suffering from coronary heart disease, angina, peripheral artery disease, and anemia [318].

Anemia may precede allergy or may be the result of allergies, as both scenarios seem to be true. People with allergies are more likely to be anemic, and the incidence of developing anemia is higher among atopic subjects. Studies report that asthmatics without anemia have a fivefold greater risk of developing anemia within 5 years [319], and 2-year-old allergic children will nearly double their risk for anemia within a year [298].

Thus, iron deficiency is common in allergic individuals [320], with allergic individuals being at greater risk for anemia. Importantly, an improved iron status has been consistently associated with a decrease in symptoms and allergic diseases.

#### 3.1.1. Iron Interventions

Iron supplementation during pregnancy, combined with folic acid, significantly reduces the risk of atopic dermatitis in children by age six [310]. A Finnish study showed that prenatal iron supplementation reduced the risk of asthma in the offspring of asthmatic mothers by nearly 70% [321]. Providing dietary iron for lymphoid uptake in pollen-allergic women improved their iron status and allergic rhinitis symptoms [30]. Oral iron supplementation for 2 months improved chronic idiopathic urticaria in patients with mild hyposideremia [244]. Clinical trials indicate that iron supplementation, rather than deworming strategies, decreases IgE levels.

#### 3.1.2. Improving the Bioavailability of Iron

A person’s health status, the form of iron, and the presence of antioxidants markedly influence the uptake of iron. Multiple uptake mechanisms for protein-bound iron, heme-iron, and nonheme iron have been described [66,174]. Vitamin C can vastly improve iron absorption, with vitamin C deficiency (scurvy) always resulting in anemia. Phytates and tannins form large complexes with iron that are not bioavailable, thus hindering its uptake [322]. Only excessively high amounts of calcium have some modest capacity to impede iron absorption [323], which is greatly improved by vitamin C [324,325] in clinical trials. Consequently, iron-fortified milk products with high calcium levels have been successfully used to improve the iron status of preterm babies [326,327], children [328], adolescents [329], and pregnant women [330]. Clinical trials have shown that in the presence of low-grade inflammation, the addition of vitamin A [329], vitamin C [331,332], and lipids [333,334] improves dietary iron uptake in inflamed settings due to rerouting absorption to the lymph.

### 3.2. Vitamin A Deficiency and Atopic Diseases

Both the insufficient and excessive intake of bioavailable vitamin A can be detrimental and lead to inflammation. Insufficient vitamin A during infancy and early childhood is associated with allergic sensitization, allergic rhinoconjunctivitis, wheezing, and food hypersensitivity [335]. Children with atopic dermatitis exhibit significantly lower serum retinol levels and impaired retinoid-mediated signaling in the skin compared with nonatopic controls [336,337], and children and adults with asthma also have lower circulating vitamin A levels [338,339,340]. As a lack of retinol is associated with increased morbidity, retinol deficiency worsens asthma [341], allergic rhinitis [342], and atopic dermatitis [277,343]. Conversely, retinol supplementation during infancy did not increase the risk of atopy at age 7 [344], and the intake of the carotenoids beta-cryptoxanthin and alpha-carotene is inversely associated with allergic skin sensitization [345]. However, persons with protein–energy malnutrition are particularly sensitive to retinol toxicity, with intakes as low as 1500 IU/kg occurring in children and pregnant women [346]. Excess intake of bioavailable lipophilic vitamin A (≥2.5 times the recommended intake of 800 RAEs/d in Nordic countries) was associated with increased asthma risk in school-age children [347]. Vitamin A deficiency potentiates Th2 inflammation and mast cell activation in atopic dermatitis [277], with an increase in serum retinol improving atopic dermatitis symptoms [278].

#### Improving the Bioavailability of Vitamin A

The bioavailability of vitamin A as a fat-soluble vitamin differs vastly, with the addition of oil being essential when consuming carotenoids for retinol uptake via the lymph [348,349]. Without oil, these carotenoids are basically not absorbed, as the conversion rate for provitamin A to retinol is 24:1, and for beta-carotene to retinol, it is approximately 12:1. [350,351,352] As such, people at risk are particularly those who consume carotenoids without added oil.

The importance of bioavailable vitamin A has been very clearly demonstrated in a prospective birth cohort study, in which supplementation of children in the first year of life with vitamins A and D in the water-soluble form doubled the risk of food allergy and asthma at the age of four compared with children receiving the same formulation in oil suspension [353]. Similarly, in children, a high intake of dietary preformed vitamin A, but not ß-carotene intake, was associated with improved lung function and lower asthma risk [354].

However, many studies do not assess the bioavailability of vitamin A, which may partly explain contradictory findings, in which some studies reported dietary beta-carotene intake reduced the risk of allergic sensitization [355,356], while another study reported an increased risk for hay fever [357]. To sum up, atopic individuals have lower retinol levels with improving vitamin A deficiency, decreasing morbidity. Intervention trials have to be interpreted carefully, taking dose and bioavailability into account.

## 4. Nutrition to Prevent Allergies

Allergic individuals suffer from numerous mineral and vitamin deficiencies [174,247,277,295,296,297,303,306,310,338,358,359,360,361,362,363,364,365,366,367,368,369,370,371,372]. Importantly, adequate levels of some trace elements such as iron are known to be crucial for adequate lung functioning [373,374], with lower serum iron levels being associated with lower lung function [301] and increased asthma [312] severity [375]. Unnecessary food avoidance is often observed in people with food allergies, increasing their risk of nutritional inadequacies. As such, food restriction has been reported to lead to micronutritional deficiencies in children with atopic dermatitis and was associated with disease severity [293,376,377]. Many allergy sufferers experience cross-allergic symptoms, reacting not only to specific proteins like PR10 or LTP in a particular allergenic source but also to cross-reactive dietary sources such as fruit and vegetables. Consequently, many allergic persons avoid a wide range of raw fruits, vegetables, and animal products out of caution. This, however, is unwarranted and may aggravate the disease course. Only specific foods that have previously caused reactions need to be avoided. When an entire food group is avoided, nutritional deficiencies are more likely to occur, further exacerbating the disease. Additionally, misconceptions about avoiding certain allergenic foods often fail to consider the stability of some allergenic proteins. For example, people allergic to cow’s milk, even if they are allergic to bovine serum albumin, can still consume cooked meat, as bovine serum albumin is thermolabile [378,379,380]. Several studies emphasize that maternal food consumption may have an impact on the development of allergic diseases, though randomized controlled trials of micronutrient supplementation do not give clear consistent information, and randomized trials focusing on food and food patterns are lacking [381]. Maternal consumption of allergenic foods such as milk and peanuts has been linked to reduced allergy and asthma risk in the offspring of a prebirth US cohort [382]. Similarly, maternal intake of fish and apples was found to be protective against the onset of asthma [383]. In the Danish National birth cohort, maternal ingestion of peanuts, tree nuts [384], and/or fish [385] decreased the risk of asthma. In contrast, maternal fish oil consumption alone did not reduce atopy [386,387,388] but did seem to slightly reduce the risk of asthma in offspring [389]. Maternal intake of vegetables and yogurt was associated with the prevention of any allergy [390], while lower maternal egg intake was linked with elevated serum total IgE and peripheral eosinophilia in children with atopic dermatitis [391]. Moreover, better iron and vitamin status [381], as well as iron supplementation during pregnancy, are associated with a lower risk of allergies and asthma in children [306,307,308,309,310,311,321]. Despite the data from current studies and systematic reviews [392], the only guidelines about maternal diet and food allergy prevention are that food allergens do not need to be avoided and that vitamin D supplementation in pregnant women with suboptimal levels may prevent offspring asthma [393,394,395].

Supporting the role of nutrient-poor conditions promoting allergy development is that numerous studies associate adequate nutrition with the prevention of atopy [30,396,397]. While not addressed in systematic reviews, several studies showed that frequent intake of nuts [382,398,399], milk [382,398,400,401], butter [398,401] wheat [382], apples [383] and other fruits [399,402,403,404], fish [383,385,399,403,405,406,407], vegetables [390,399,402], yogurt [390], and meat (of high quality) [404,407] in childhood is associated with reduced allergy and asthma risk in children. Meta-analyses revealed that probiotics alone or combined with prebiotics can reduce atopic dermatitis symptoms in children without food allergies [408].

In the Spanish ISAAC phase III, the intake of cow’s milk, butter, and nuts was found to reduce the risk of atopic dermatitis in children [398]. In the GABRIELA cohort, raw cow’s milk consumption was associated with a reduced risk of asthma and atopy, with whey protein levels being inversely associated with asthma [400].

One-month consumption of a whey-based oral supplement was able to reduce total IgE levels and improve lung function in asthmatic children [409]. In a randomized controlled trial from Brazil, consumption of a micronutrient and a prebiotic-fortified milk beverage for 6 months decreased the risk of allergic manifestations by 36% [410]. Moreover, consumption of a whey supplement fortified with iron, vitamin A, and zinc for 3–6 months ameliorated symptoms of allergic rhinitis [30,411,412]. Additionally, drinking raw milk was better tolerated in allergic children than highly processed milk in a pilot study [413].

## 5. Conclusions

Allergic individuals are at increased risk of malnutrition, particularly with deficiencies of iron and vitamin A, compared with those without allergies. These nutrient deficiencies have a profound impact, triggering nutritional immunity, type 2 inflammation, and restricting dietary uptake of micronutrients. Allergenic proteins can bind nutrients very effectively and are able to evoke nutritional immunity in persons with protein or micronutritional deficiencies. However, these same proteins in nutrient-adequate conditions act as carriers for micronutrients and contribute to immune health. Despite the high prevalence of malnutrition and the impact of protein and micronutrient deficiencies on the development as well as the severity of allergic diseases, nutritional care is many times inadequate, with many in the medical/nursing field not being able to diagnose malnutrition and not including dietary measurements to prevent the disease course.

Nutritional education for both people with allergies and health care professionals plays a crucial role in preventing the atopic march. Consumption of allergenic food such as milk, whey products, fish, nuts, fruits, and vegetables should be encouraged, as these foods are rich in micronutrients and have been shown to be beneficial for the prevention and amelioration of the atopic state. Malnutrition can be prevented through nutritional education and the consumption of a healthy, varied diet, as well as by fortifying foods or direct supplementation as needed.

## Figures and Tables

**Figure 1 jcm-13-04713-f001:**
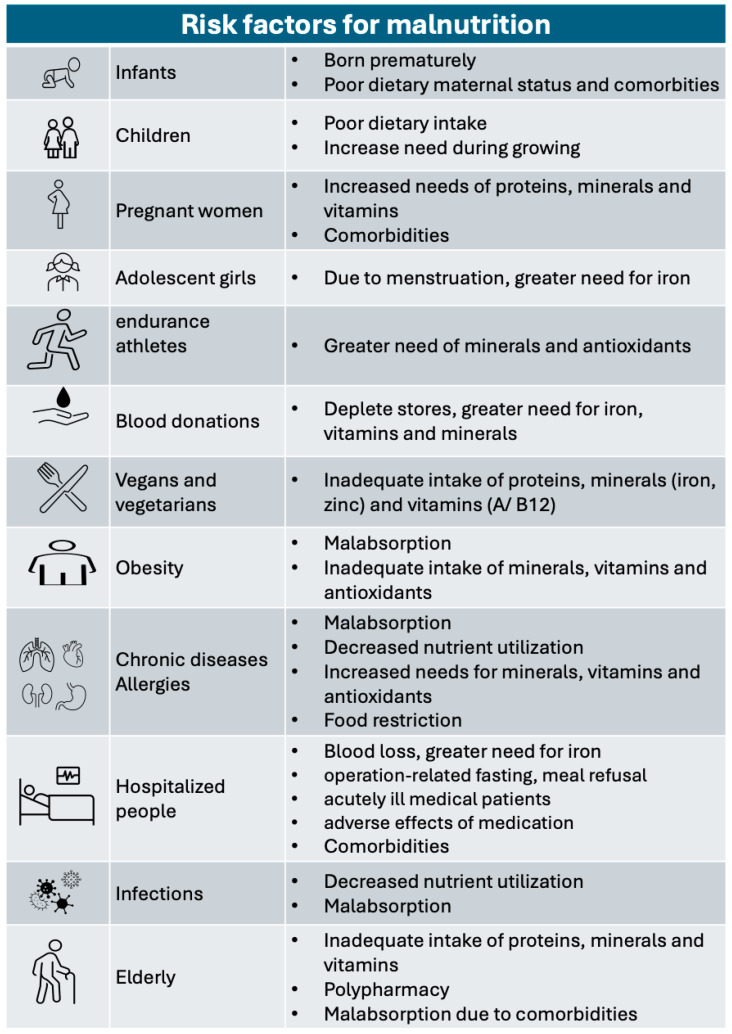
Risk factors for malnutrition.

**Figure 2 jcm-13-04713-f002:**
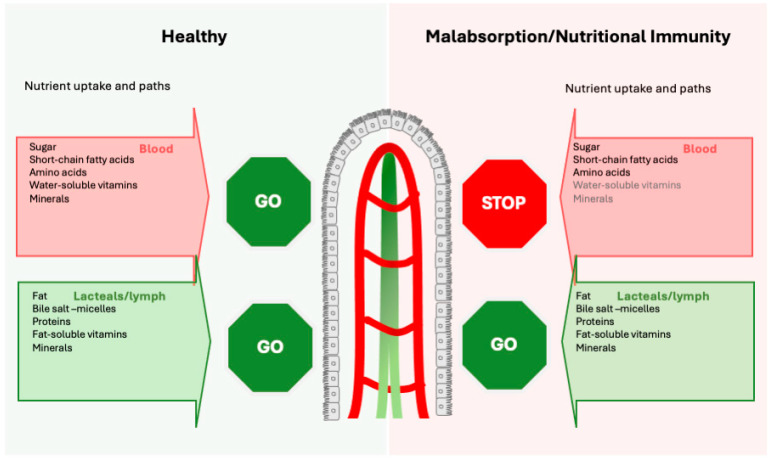
Nutritional Immunity promotes malabsorption. While in the normal steady state, water-and fat-soluble compounds cross the epithelial barrier and enter the body via the blood system and/or the lacteals, inflammation will trigger nutritional immunity. This results in impaired absorption of minerals and vitamins, particularly in those following the blood route. In contrast the “lymph route” remains accessible as it still allows monitoring of nutrients for potential pathogens.

**Figure 3 jcm-13-04713-f003:**
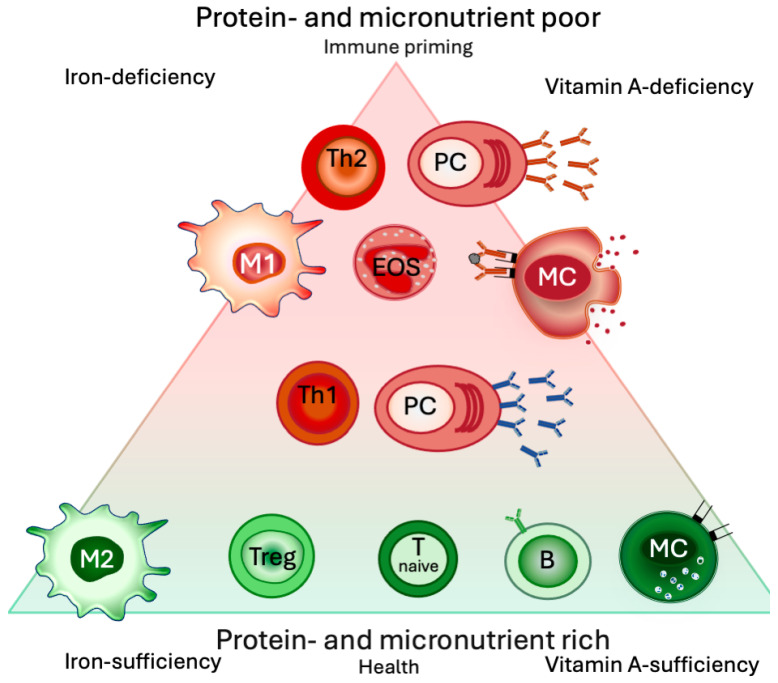
Protein- and micronutrient-poor conditions promote type 2 inflammation. Micronutrientrich conditions foster a regulatory and anti-inflammatory phenotype in lymphocytes, macrophages, and mast cells, while nutrient-poor conditions prime the immune system. A lack of micronutrients, particularly of iron and vitamin A, initially mounts a Th1/Th17-dominated immune response, which results in B cells transforming into plasma cells and secreting IgG-antibodies. When nutrient-poor conditions persevere for longer time periods, the immune response shifts toward Th2 (due to the more nutrient-sensitive nature of Th1 cells) and promotes eosinophils, as well as class switch toward IgE antibodies. M2: regulatory macrophage, Treg: regulatory T cells, B: naïve B-cells, EOS: eosinophils, MC: mast cells, PC: plasma cells.

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
