# Peer review of "Malnutrition and Allergies: Tipping the Immune Balance towards Health"

_jcm, 2024, doi:10.3390/jcm13164713_

Round 1

Reviewer 1 Report

Comments and Suggestions for Authors

The manuscript is well written and clearly presented. The topic is interesting. It is well documented. The graphs are very nice and explanatory.

Add to Table 1 patients admitted to hospital wards, especially in surgical wards, and mention this concept in the text.

Expand the paragraph discussing malnutrition in allergic diseases by also specifying that cross-reactivity phenomena and misconceptions can lead to incorrect dietary indications. For example, the belief that it is not appropriate to eat beef in patients allergic to cow's milk is still very common today and does not take into account that this cross-reaction is due to sensitization to bovine serum albumin, which can be tested, and that this protein is thermolabile and therefore does not preclude the consumption of cooked meat. Even more complex, especially in the Mediterranean area, is sensitization to nsLTP, which often leads patients, if not well supported by the doctor, to drastically reduce the intake of vegetables, with significant consequent nutritional deficiencies.

Author Response

Comment 1: The manuscript is well written and clearly presented. The topic is interesting. It is well documented. The graphs are very nice and explanatory.

Response 1: We appreciate your overall positive feedback.

Comment 2: Add to Table 1 patients admitted to hospital wards, especially in surgical wards, and mention this concept in the text.

Response 2: Thank you. We now added in the table “hospitalized people” with the associated risk factors and added in the text a short sentence on hospital acquired malnutrition with some major risk factors and hope thereby to meet the reviewer’s demands adding following sentences:

Reasons for hospital-acquired malnutrition is beside poor appetite, hospital meal refusal, operation-related fasting, also polypharmacy and comorbidities [7], with malnutrition being an predictor of post-operative morbidity and mortality[8].

Comment 3: Expand the paragraph discussing malnutrition in allergic diseases by also specifying that cross-reactivity phenomena and misconceptions can lead to incorrect dietary indications. For example, the belief that it is not appropriate to eat beef in patients allergic to cow's milk is still very common today and does not take into account that this cross-reaction is due to sensitization to bovine serum albumin, which can be tested, and that this protein is thermolabile and therefore does not preclude the consumption of cooked meat. Even more complex, especially in the Mediterranean area, is sensitization to nsLTP, which often leads patients, if not well supported by the doctor, to drastically reduce the intake of vegetables, with significant consequent nutritional deficiencies.

Response 3: Thank you very much for addressing this very important aspect particularly in food allergy, in which avoidance of whole food categories due to fear of allergic reactions is a major problem aggravating the disease course. To address this very present misconception in allergic people, we now added the following paragraph in the section “4. Nutrition to prevent allergies”: 

Unnecessary food avoidance is often observed in people with food allergies, increasing their risk of nutritional inadequacies. As such, food restriction has been reported to lead to micronutritional deficiencies in children with atopic dermatitis and was associated with disease severity[302, 386, 387]. Many allergy sufferers experience cross-allergic symp-toms, reacting not only to specific proteins like PR10 or LTP in a particular allergenic source, but also to cross-reactive dietary sources such as fruit and vegetables. Conse-quently, many allergic persons avoid a wide range of raw fruits, vegetables and animal products out of caution. This, however, is unwarranted and may aggravate the disease course. Only specific foods that have previously caused reactions need to be avoided. When an entire food group is avoided, nutritional deficiencies are more likely to occur, further exacerbating the disease. Additionally, misconceptions about avoiding certain allergenic foods often fail to consider the stability of some allergenic proteins. For example, people allergic to cow’s milk, even if they are allergic to bovine serum albumin, can still consume cooked meat as bovine serum albumin is thermolabile[388-390].

Thanks for the very constructive suggestion.

Reviewer 2 Report

Comments and Suggestions for Authors

With real interest, I read the manuscript jcm-3140523. It is a monumental work, comprehensively addressing a very important topic. And it is clear , that it is written by renowned researchers, experts in the field.

I have minor comments only:

1.      Figure 1. “SCFA”. It is never explained in the legend or in the main text what it is. Please, explain this abbreviation and maybe drop a line on SCFA somewhere in the text.

2.      Details. E.g. line 249: why “Iron” not “iron”. Table I: sometimes you start with the upper and sometimes with the lower case. Line 200: why “Ara h 2” but “Arah6”? Etc.

3.      Btw., Table I does not have a title and a legend.

4.      Recently, it is more common to talk about “type 2 (1)” not “Th2 (1)” inflammation, milieu, immunity, etc. (PMID: 30057383). The Authors are not uniform in the article.

5.      Language. Small improvements in very rare cases are required, e.g. line 120: “this result”.

6.      Line 176: “IL4/Ifngamma”. The first is human, the second murine. I would try to stay as conservative as possible, e.g. murine nomenclature for mice, and human for humans. Or I would try to unify the things throughout the manuscript by using only human nomenclature (and clearly stating it).

7.      The last sentence and at the same time one-third of the conclusions (lines 560-563): is this statement proportional (e.g. in size) to the chapter justifying it in the main text (lines 173-232)?

8.      Further perspectives and future directions?

Comments on the Quality of English Language

Small improvements in very rare cases are required.

Author Response

Comment 1: With real interest, I read the manuscript jcm-3140523. It is a monumental work, comprehensively addressing a very important topic. And it is clear , that it is written by renowned researchers, experts in the field.

Response 1: We highly appreciate your very positive feedback.  

Comment 2: I have minor comments only:

Figure 1. “SCFA”. It is never explained in the legend or in the main text what it is. Please, explain this abbreviation and maybe drop a line on SCFA somewhere in the text.

Re

Response 2: SCFA stands for short-chain fatty acids, which we changed in the figure and the abbreviations now also mentioned in the text.

Comment 3:

Details. E.g. line 249: why “Iron” not “iron”. Table I: sometimes you start with the upper and sometimes with the lower case. Line 200: why “Ara h 2” but “Arah6”? Etc.

Response 3: We appreciate your detailed revisions. We have revised our manuscript, and hopefully remove all grammar and spelling mistakes.

Comment 4:Btw., Table I does not have a title and a legend.

Response 4: We now added a title and a very short legend.

Comment 5: Recently, it is more common to talk about “type 2 (1)” not “Th2 (1)” inflammation, milieu, immunity, etc. (PMID: 30057383). The Authors are not uniform in the article.

Response 5: Absolutely true, we substituted the terminus and included the reference to the high BMI and asthma section.

Comment 6:

Language. Small improvements in very rare cases are required, e.g. line 120: “this result”.

Response 6: Thank you, we proof-read the whole manuscript and corrected for grammatical and spelling errors.

Comment 7:

Line 176: “IL4/Ifngamma”. The first is human, the second murine. I would try to stay as conservative as possible, e.g. murine nomenclature for mice, and human for humans. Or I would try to unify the things throughout the manuscript by using only human nomenclature (and clearly stating it).

Response 7: Thank you, we now throughout the manuscript use the human nomenclature.

Comment 8:

The last sentence and at the same time one-third of the conclusions (lines 560-563): is this statement proportional (e.g. in size) to the chapter justifying it in the main text (lines 173-232)?

Further perspectives and future directions?

Response 8: Again thank you for the very valid comment, we elaborated the importance of allergenic protein further to give them more justice in the conclusion section

We also added some future perspectives to emphasize to importance to address malnutrition in atopic individuals

Following sentences were added in the conclusion section

Allergenic proteins can bind nutrient very effectively and are able to evoke nutritional immunity in persons with protein- or micronutritional deficiencies. However, these same proteins will in nutrient-adequate conditions act as carriers for micronutrients and con-tribute to immune health. Despite the high prevalence of malnutrition and the impact of protein- and micronutrient deficiencies on the development as well as severity of allergic diseases , nutritional care is many times inadequate, with many in the medical/ nursing field not able to diagnose malnutrition and not including dietary measurements to prevent the disease course.

Nutritional education for both people with allergies and healthcare professionals plays a crucial role in preventing the atopic march.. 

We hope thereby that all queries of the reviewer could be addressed.